# Learning to learn with better convergence

## Abstract

We consider the learning to learn problem, where the goal is to leverage deep learning models to automatically learn (iterative) optimization algorithms for training machine learning models. A natural way to tackle this problem is to replace the human-designed optimizer by an LSTM network and train the parameters on some simple optimization problems (Andrychowicz et al., 2016). Despite their success compared to traditional optimizers such as SGD on a short horizon, these learnt (meta-) optimizers suffer from two key deficiencies: they fail to converge (or can even diverge) on a longer horizon (e.g., 10000 steps). They also often fail to generalize to new tasks. To address the convergence problem, we rethink the architecture design of the meta-optimizer and develop an embarrassingly simple, yet powerful form of meta-optimizers—a coordinate-wise RNN model. We provide insights into the problems with the previous designs of each component and re-design our SimpleOptimizer to resolve those issues. Furthermore, we propose a new mechanism to allow information sharing between coordinates which enables the meta-optimizer to exploit second-order information with negligible overhead. With these designs, our proposed SimpleOptimizer outperforms previous meta-optimizers and can successfully converge to optimal solutions in the long run. Furthermore, our empirical results show that these benefits can be obtained with much smaller models compared to the previous ones.

## 1 Introduction

Optimization is an important problem for almost all tasks in machine learning. For optimizing non-convex problems that arise in machine learning, such as minimizing training loss of a neural network, SGD is still the de-facto algorithm. However, success of SGD often hinges on careful selection of learning rate and its decay schedule. But finding such a schedule for a new problem is laborious, time-consuming and computationally expensive. To resolve this issue, adaptive methods such as AdaGrad (Duchi et al., 2011) and ADAM (Kingma & Ba, 2014) were proposed. These methods adaptively re-scale the step-size and typically do not require such a learning rate schedule. However, they still require tuning of initial learning rate, and more importantly, can have worse generalization compared to SGD (Wilson et al., 2017; Berrada et al., 2018).

Recently, learning to learn paradigm (L2L) was proposed to tackle this problem. This approach attempts to eliminate the need for human-designed optimization rules in favor of a meta-learner capable of learning a good optimization update strategy from experience (Lv et al., 2017). Here, experience refers to other related optimization tasks. Recurrent neural network (RNN) is the common choice for the meta-optimizer since it can capture long term dependencies, which is of paramount importance in optimization. Recent results show that a well-trained, LSTM-based meta-optimizers often outperform hand-tuned approaches on supervised tasks (Andrychowicz et al., 2016; Wichrowska et al., 2017).

Despite the promising initial results in this field, previous work on meta-optimizers often struggle to generalize to new problems. More importantly, these meta-optimizers usually cease to make progress (and can sometimes even diverge) after running for a large number of steps. To handle this issue, existing methods either add many pre-processing rules inspired by a traditional optimizer (Lv et al., 2017) or employ very complicated architectures (Wichrowska et al., 2017), often leading to reproducibility issues. Furthermore, none of these methods clearly resolve the issue of generalization in a longer run. To make matters worse, the complex architectural design of the existing meta-optimizers often hinders attempts to understand the issue from technical angle.

In the light of these problems, we rethink the architectural design of the meta-optimizer and design a simple yet clean form of meta-optimizers—a coordinate-wise RNN model with embedding sharing. We then carefully analyze the problems with the previous design and fix those problems by re-designing its structure and loss, thereby, obtaining a new meta-optimizer with much better generalization properties. More specifically, our contributions are as follows:

- **Structure of each RNN:** We provide a careful analysis of vanilla RNN-based meta-optimizer, and identify subtle issues related to the bias term in the meta-optimizer. We empirically show that by removing the bias term in RNN, long horizon convergence issues can be addressed.

- **Loss function:** We show the loss function used in previous L2L frameworks tend to favor a myopia meta-optimizer, one that quickly decreases the objective in a short term but has slow long-term convergence. Inspired by the convergence rate of SGD, we propose a new loss based on weighted difference, which helps the meta-optimizer to explore a better strategy in the later stage of optimization.

- **Information sharing betwen RNNs:** Motivated by second-order optimization methods, we propose a novel embedding sharing technique to enable information exchanging between the RNN of different coordinates. This new technique successfully improves the convergence speed of meta-optimizers by sharing information across coordinates.

- **Fine-tuning scheme:** We propose a novel strategy to fine-tune the meta-optimizer. In this strategy, when given a new task, the learnt meta-optimizer can be quickly fine-tuned on a small subset of data. Experimental results show that this scheme works well in practice. This will enable the meta-optimizer to seamlessly generalize to a wider set of optimization problems, ones which it did not encounter during training.

With these design changes, our proposed SimpleOptimizer converges much faster than the previous meta-optimizers and can successfully converge to optimal solutions in the long run. Furthermore, these benefits can be reaped at no additional cost of model complexity; in fact, the number of parameters in our model is smaller than the previous works. For instance, our optimizer (in Section 4) only has 132 parameters, while the previous meta-optimizers have at least 5000 parameters, with per-iteration speed 10 times slower than our method. This property is of great importance when meta-optimizer is applied to large models.

## 2 RELATED WORK

The area of meta-learning has garnered significant interest recently (Brazdil et al., 2008), and includes topics such as hyper-parameter tuning (Golovin et al., 2017; Thornton et al., 2013) and architecture search (Zoph & Le, 2016; Liu et al., 2018). We mainly focus on learning a meta-optimizer to optimize loss functions for training machine learning models. In this section, we briefly review current training and meta-learning algorithms that are most relevant to our work.

**Adaptive Gradient Methods** Stochastic gradient descent (SGD) has been commonly used in training neural networks. At each step, SGD picks a batch of training samples, computes gradient of the batch and then conducts model updates. Several techniques have been proposed to improve SGD and eliminate the need to hand-tune its learning rate. This includes momentum updates (Sutskever et al., 2013), Adagrad (Duchi et al., 2011), which uses an adaptive learning rate for each dimension, and ADAM (Kingma & Ba, 2014) which combines adaptive rules and momentum updates. There are several recent discussions about Adam's convergence behavior and improved version of Adam (Reddi et al., 2018). Also, a few recent works show that SGD with a carefully chosen learning rate can be competitive or even outperform Adam (e.g. Wilson et al. (2017)).

**Learning step size** Some of meta-learning works aim to learn the learning rate schedule directly. In these works (Baydin et al., 2017; Wu et al., 2018; Li & Malik, 2016), gradient information is used as input to meta-optimizer, but update direction is still aligned with gradient direction. In general, not all of these methods employ RNNs as a meta-optimizer. For example, in Li & Malik (2016), the learning problem is framed as an MDP problem, which can then be naturally tied to reinforcement learning. While this line of research falls under the umbrella of meta-learning, our paper focuses on

finding a meta-optimizer generating both learning rate as well as direction updates. Thus, we do not include this line of research in our empirical comparisons.

**Learning to learn (L2L)**   Instead of using some human-defined update rules as optimizers, a natural question to ask is whether it's possible to automatically learn the update rule. The seminal work by Cotter & Conwell (1990) argued that RNN can be used to model optimization algorithms, and some earlier attempts have been made in Younger et al. (2001) to use gradient descent to learn a RNN optimizer on convex problems. Recently, due to the difficulty of optimizing deep networks, the problem of learning meta-optimizers has become an important topic. Andrychowicz et al. (2016) proposed to use LSTM to model the update rule of a meta-optimizer, and showed that the learnt rules on simple problems can generalize to more complex problems. However, it is also reported in the paper that the learnt optimizer cannot generalize to long-term training—the algorithm cannot converge to stationary points when running for a longer horizon. To resolve this issue, Lv et al. (2017) proposed several modifications, including randomly scaling the parameters of the model and pre-processing the gradients to mimic ADAM, and a new architecture based on ELU (Exponential Linear Unit) (Clevert et al., 2015). Wichrowska et al. (2017) proposed multi-layer hierarchical LSTMs as the meta-optimizer, extending the input dimension by adding momentum, dynamic scaling and attention mechanism.

Unfortunately, even with the sophisticated design of meta-optimizer, the generalization issues (to more steps or various models/datasets) are not fully resolved. Lv et al. (2017) fails to generalize to longer than 10K steps and Wichrowska et al. (2017) fails to generalize to deeper model such as GoogleNet. Although pre-processing (e.g., like equation equation 3) is claimed to be important in all the previous works, there is no clear reason given for that. Furthermore, the architecture in Wichrowska et al. (2017) is complicated and hard to validate. Most importantly, there is still very little understanding about the root cause of the unstable convergence behavior.

## 3   PROPOSED META OPTIMIZER

In this section, we first define the notation used in the paper and introduce the overall architecture of our meta-optimizer. *Optimizee $f$* is the task or loss function we want to minimize. At each time step, a random batch of training data is sampled and the optimizee model $f$ will use the data to generate a gradient, which is then fed as an input to a *meta-optimizer $m$*. While the Optimizer $m$ could be any model, in this paper, we restrict our attention to the case where $m$ is a recurrent neural network (RNN). To make the optimizer generalizable to problems with various dimensions, the whole procedure is done in a **coordinate-wise manner**. Each coordinate of $f$ will maintain a separate hidden state and share the same RNN parameters. Thus, the update rule of optimizing $f$ can be written as:

$$\theta_{t+1} = \theta_t + m(g_t), \tag{1}$$

where $\theta_t$ denotes the parameters of optimizee model $f$ at time step $t$, and $g_t$ is the gradient of $f$ based on the sampled batch of training data at step $t$. Training of RNN requires us to select a truncated Backpropagation Through Time (BPTT) steps. Within a time horizon $T$, each hidden state of RNN will relate to future hidden state generation and it allows the model to capture longer dependencies in the optimizee task. The objective function of training the meta-optimizer for some horizon $T$ steps can be written as:

$$L(m) = \sum_{t=1}^{T} w_t f(\theta_t). \tag{2}$$

Here $w_t$'s are arbitrary weights associated with each time-step. Our overall framework is illustrated in Figure 1. We defer the discussion on "extract information share" in Figure 1, which is a new structure for information sharing between coordinates, to Section 3.3.

### 3.1   STRUCTURE OF EACH RECURRENT UNIT

**Previous design**   In the first L2L paper (Andrychowicz et al., 2016), the authors proposed to use LSTM for each recurrent unit. To stabilize the meta-optimizer's convergence, they further proposed to pre-process gradient by the following function before feeding into LSTM:

$$f(g) = \begin{cases} (\frac{\log |g|}{p}, \mathrm{sgn}(g)) & \text{if } |g| \geq e^{-p} \\ (-1, e^p g) & \text{else,} \end{cases} \tag{3}$$

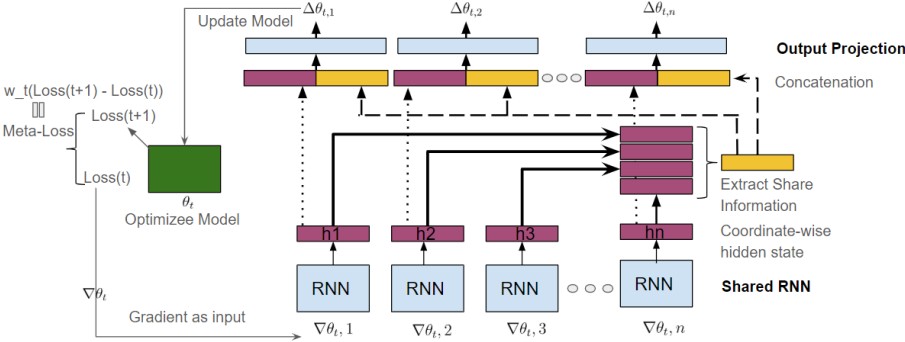

Figure 1: Illustration of the proposed SimpleOptimizer architecture.

where $p$ is a parameter controlling how small gradients are disregarded. All followup works build upon a similar LSTM preprocessing structure.

**Our design** Instead of simply using a LSTM, we take a step back and study characteristics of the recurrent cell that leads to better convergence. To this end, it is instructive to take the meta-optimizer $m$ to be vanilla RNN with only 2 hidden states to demonstrate the subtle problem with the bias term. In this setting, the update rule of equation 1 becomes:

$$\theta_{t+1} = \theta_t + \tanh(w_1 g_t + w_2 h_t + b) \tag{4}$$
$$h_{t+1} = \tanh(w_1 g_t + w_2 h_t + b), \tag{5}$$

where $w_1, w_2$ are the RNN weights and $b$ is the bias term. As we optimize function $f$ by changing $\theta$, we will gradually get smaller loss and, thus, smaller gradients. When $\theta$ approaches a stationary point of the function $f$, in order to converge, the update needs to be small enough to stay close to the stationary point. However, from the update rule as in equation 4 and equation 5, we can clearly see that even when $g_t$ is small, unless $b$ equals to $0$, the update rule will still produce an update with magnitude proportional to $\tanh(b)$. In the extreme case when $\theta$ is the optimal point and when the previous hidden state is almost $0$, this update rule will still pushes the parameter away from the optimal point. Thus, the bias term $b$ is problematic as optimization proceeds close to the optimal solution.

The same observation can be made even with the pre-processing methods equation 3. For these methods, depending on the gradient magnitude, we could further write the update as:

$$\theta_{t+1} = \theta_t + \tanh(w_{11}\frac{\log|g_t|}{p} + w_{12}\text{sign}(g_t) + w_2 h_t + b),$$
$$\text{or} \quad \theta_{t+1} = \theta_t + \tanh(-1 w_{11} + w_{12} e^p g + w_2 h_t + b).$$

Notice the constant terms 1 or -1 in the equations. This, again, will be problematic when the corresponding parameters $w_{11}$ and $w_{12}$ are not 0. Clearly, pre-processing involving the separation of the magnitude and the direction of gradients does not help to address the problem. This is an issue in practice as well. For instance, in one of our experiments, we notice that the maximal bias term of LSTM in Lv et al. (2017) is 0.41, and maximal bias term of LSTM in Andrychowicz et al. (2016) is 0.31; this, consequently, leads to bad performance. Meta-optimizers fail to learn bias$= 0$ since they are only trained in a limited horizon and it is beneficial to "not" have bias term to be 0 to speed up initial stage of optimization.

Following the analysis, we conclude that the bias term in the meta-optimizer can cause convergence issues. If we remove the bias term in the RNN model, we hypothesize that we do not require any pre-processing to ensure that the model trains well and behaves properly when applied on a longer horizon. We empirically validate this statement in the later part of the paper (see Section 4).

### 3.2 WEIGHTED DIFFERENCE OF OBJECTIVE FUNCTIONS

Another problem in previous learning-to-learn algorithms is the design of loss function.

**Previous loss function**    For the previous meta-optimizer's loss function, weights in  equation 2 are either set to 1 or 0. For example, in Andrychowicz et al. (2016), $w_t$ is set to 1 for all steps ($\sum_{t=1}^{T} f(\theta_t)$) and this corresponds to requiring meta-optimizer to generate better loss for all T steps. In Lv et al. (2017), authors propose to set $w_t$ to be 0 for all t except $w_T$ to be 1 for the final step. Thus, the loss only depends on $f(\theta_T)$, and when running for $iT$ iterations the loss will be $f(\theta_T) + f(\theta_{2T}) + \cdots + f(\theta_{iT})$.

We first discuss why the previous loss function leads to a myopia meta-optimizer that has slower convergence in the long term. In Andrychowicz et al. (2016) the weights are uniform ($w_t = 1$ for all $t$). Consider the following two sequence of loss ($f(\theta_t)$) under 6 BPTT steps:

$$\text{loss of sequence 1 :} 100, 20, 5, 1.25, 0.625$$
$$\text{loss of sequence 2 :} 100, 2.5, 2.25, 2.125, 2.3125$$

in the uniform weighting scheme, the loss of sequence 1 is larger than sequence 2, so a meta-optimizer that produces sequence 2 is preferred. However, in terms of optimizing a function for a longer run, clearly sequence 1 is a better choice. In Lv et al. (2017), the loss will be $f(\theta_T) + f(\theta_{2T}) + \cdots + f(\theta_{iT})$, which will still face the same problem—a sequence that reduces objective function faster in the beginning is preferred over a sequence that decreases slower but could generalize to more steps.

**Our proposed loss function**    Intuitively, we want to have higher weights for larger $t$ to avoid meta-optimizer shortsightedly focusing on initial stage of learning, but how to set up the weights? We propose a new weighted loss motivated by the convergence properties of SGD. For a strongly convex problem, it is known that SGD has an $O(1/T)$ convergence rate, which means

$$f(\theta_t) - f(\theta^*) \approx C/t,$$

where $C$ is a constant. However, in practice we do not know $f(\theta^*)$, and to avoid that we consider

$$f(\theta_{t-1}) - f(\theta_t) \approx f(\theta^*) + C/(t-1) - f(\theta^*) - C/t = C(\frac{1}{t-1} - \frac{1}{t}) \approx O(\frac{1}{t^2}).$$

This means that the difference between consecutive losses will decrease quadratically. To make all the updates equally important in the final loss function of meta-optimizer, we should compensate the smaller difference of losses by giving a larger weight (e.g., $O(t^2)$) to the objective function reduction at the $t$-th step. The rate can be changed due to different convergence properties of the meta-optimizer, but it should always be an increasing sequence to make the weighted difference of losses become roughtly euqal.

Therefore, we use the following objective function to train meta-optimizer:

$$L(m) = \sum_{i=1}^{T} w_i(f(\theta_{i-1}) - f(\theta_i)), \tag{6}$$

where $w_i$ is an increasing function ($O(i^2)$) of $i$, which means that more weightage is given to the later stage of optimization. Without this weighting function, the meta-optimizer will tend to focus only on the initial stage of optimization.

### 3.3    Embedding sharing between coordinates

**Previous design.**    A meta-optimizer uses a coordinate-wise recurrent structure to adapt to problems with any dimension. Thus, for a $d$-dimensional problem, most of previous works consider meta-optimizers with $d$ independent chains, and there is no information exchange between coordinates of the update. Notice that some prior works have already designed certain information exchange mechanism, but it's either in another line of research (Li & Malik, 2016) or too complicated to validate (Wichrowska et al., 2017). In our design, we will demonstrate a simple yet effective mechanism could be achieved.

**Our design.**    In order to make a meta-optimizer learn to exploit second order information, it is necessary to design a mechanism to enable information sharing between those $d$ RNN chains. We propose a novel embedding sharing mechanism to achieve this goal. Let $h_1, \ldots, h_d$ be the hidden states of those $d$ RNN chains, we use an aggregator to gather information from all these hidden states

into an embedding vector, and then concatenate this new hidden state with the original hidden state of each RNN chain. This can be formulated as

$$h_t \leftarrow [\alpha \cdot h_t, \ \beta \cdot \text{Aggregate}(\{h_i\}_{i=1}^d)], \ \ \forall t = 1, \dots, d,$$

where $\alpha, \beta \in R$ are two learnable variables shared by all the coordinates (in order to generalize to arbitrary dimension), and Aggregate is the aggregation function that gathers an arbitrary number of embeddings into a fixed sized vector. In general, any aggregation function can be applied. In our experiments we set it to be simple mean, which consistently yields better results than running $d$ independent RNN chains. Each $h_t$ will then be used to compute the update vector and meanwhile passed to the next iteration, so the global information can affect all the RNN chains. Furthermore, this aggregation mechanism only has two parameters $\alpha, \beta$ and is thus, easy to train. We show in the experiments that this simple information sharing mechanism leads to further improvement, enabling the meta-optimizer to outperform standard optimizers like SGD and ADAM.

### 3.4 FINE-TUNING

The current L2L frameworks train the meta-optimizer on one or a set of "basic problems" and hope the optimizer can generalize to different types of neural networks Wichrowska et al. (2017). Emprically, it is observed that the learned optimizer does not work well on all types of networks. In addition, as neural network designs include more and more diverse types of architecture, it's difficult to define what subset of problems are enough to cover all different training dynamics in real world problems.

To solve this problem we propose another scheme when the meta-optimizer is applied to new problems. We introduce a fast fine-tuning training step in order to let meta-optimizer slightly adjust the parameters. Each time when a new optimizee model or dataset comes, we firstly sub-sample 5% of the training data and then fine-tune the meta-optimizer on it. Since the sub-sampled dataset is smaller than the full training set, the training speed would be much faster than the exploring on full dataset. At each iteration, meta-optimizer keeps taking input batches and update the optimizee for certain fixed steps until the average training loss is less than 1/2 of initial sampled training loss. We repeat this procedure for 5 times and stop the warm-up training. This fined-tuned meta-optimizer is then applied to train the optimizee on the full dataset. We empirically show that this procedure does not make the meta-optimizer overfit to the sub-sampled small dataset and it leads to better training loss on the full dataset. Note that fine-tuning introduces only ignorable overhead—usually less than 1000 batches are used for fine-tuning and this enables much better generalization to 10K or more steps. Furthermore, empirically we found the meta-optimizer is not sensitive to the sub-sample rate for fine-tuning. On all the problems we tested, any numter between 3% and 5% leads to good results.

## 4 EXPERIMENTS

In all the experiments, we firstly train meta-optimizers on a 2 layer MLP on MNIST dataset. The first layer maps the input into 500 dimensions and the second layer further projects it into logits of 10 classes. ReLu is used as the activation function in the neural network. We follow the literature to call this model base MLP. To demonstrate the effectiveness of the proposed method, we just use a vanilla RNN with 10 hidden dimensions as the meta-optimizer. An additional linear layer of size 20x1 (10 for hidden state and 10 for aggregate vector) is applied in order to perform coordinate-wise update. Most importantly, we disable bias term of both RNN and linear output. There is no further pre-processing, so the coordinate-wise gradient is directly fed into the vanilla RNN. We use ADAM optimizer with lr = 1e-3 to train meta-optimizers. We provide code submission to implementation of Baseline MLP task and complete code will be released on submission to ensure the reproducibility.

We compare the proposed method (SimpleOptimizer) to two baseline methods. The first method is the primitive meta-optimizer introduced in Andrychowicz et al. (2016). It contains 2 layers of LSTMs and the hidden dimension is set to 20. We call this model DMOptimizer. The other meta-optimizer is RNNPROP Lv et al. (2017). It adds ADAM-like pre-processing units and adopts sampled-scaling of optimizee functions in every training step. In Table 1, we list the model sizes of the three meta-optimizers. Clearly our model is much smaller and simpler than the previous approaches, yet, as we shall see it is able to generalize better than the previous ones. Furthermore, due to smaller size, our model enjoys much faster (more than 10 times) computation time, which means limited overhead added over gradient computation for each step.

Table 1: Number of parameters and inference speed of three meta-optimizers.

|  | SimpleOptimizer | DMOptimizer | RNNProp |
|---|---|---|---|
| # parameters | 132 | 5301 | 6801 |
| Time per step (ms) | 1.96 | 21.88 | 37.02 |

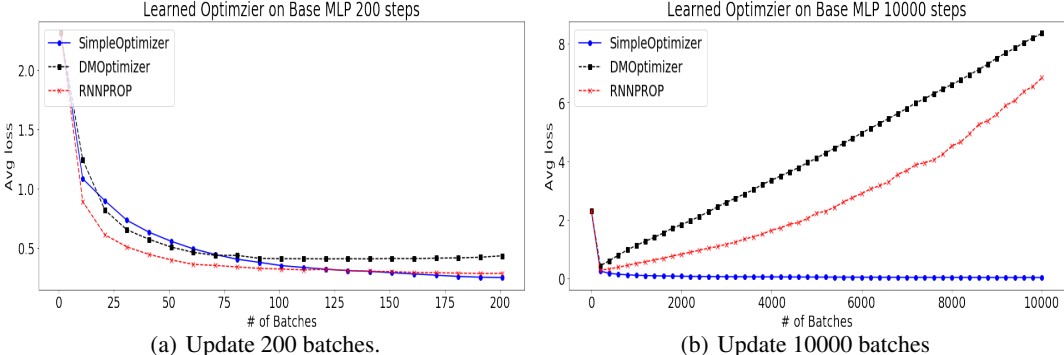

(a) Update 200 batches.

(b) Update 10000 batches

Figure 2: Comparison to baseline models. Results show that baseline models still have convergence issue over 10k steps. In contrast, SimpleOptimizer resolves the issue without complicated design.

In addition, we also compare meta-optimizers with standard optimizer such as SGD and adaptive optimizer such as ADAM. When evaluating these methods for basic setup, we choose a fixed learning rate by searching for the best performing one from the set $\{1, 0.1, 0.01, 0.001\}$ and report the best performance. When evaluating under the fine-tuning setup, we have both meta-optimizer and SGD/ADAM to train/search the best learning rate in the small sub-sampled dataset. The best setup found will then be used in the final evaluation over whole dataset.

**Comparison to Baseline Methods**   We first compare our SimpleOptimizer with the baselines. Following the same setting as the previous work Andrychowicz et al. (2016), we train all meta-optimizers on base MLP with 20 BPTT steps and train on a model for only 200 steps in total. Trained meta-optimizers are evaluated on the same setup except they are applied in up to 10,000 steps. The dataset used is MNIST. In Figure 2(a), we can observe that RNNPROP indeed improves over the vanilla L2L model, and leads to a decent decrease in loss over initial 200 steps; however, as shown in figure 2(b), when applying it with significantly many steps (over 200 steps), it does not converge. On the other hand, results show that our approach leads to very good convergence. The model keeps decreasing toward zero even after 10,000 batches of data.

**Ablation Analysis of Proposed Method**   We conduct ablation analysis on the proposed SimpleOptimizer to see the effect of each proposed idea. Each time, we remove one proposed feature from the model and compare the performance. In Figure 3(a), we train an optimizer by adding back the bias term. We can observe that this model cannot approach a better solution. Previous works seek to address this by having more complicated models (e.g. LSTMs) whereas we point out that the root cause of the problem is the presence of the bias term. In Figure 3(b), we compare SimpleOptimizer versus the same architecture without aggregated information. We can observe that sharing information between coordinates leads to faster convergence. Also, by comparing SimpleOptimizer with and without weighted loss, we conclude that weighted loss is important for convergence.

**Generalization to Convolutional Neural Network (CNN)**   In this experiment, we check whether the meta-optimizer trained on the base MLP could generalize its capability to another architecture. In particular, we want to see if our meta-optimizer (trained on base MLP) could perform well on a convolutional neural network, more specifically, LeNet. As shown in Figure 4(a), the proposed method can still optimize CNN very well, and it even outperforms a well-tuned SGD. This could imply that given the same dataset, the meta-optimizer can be transferred to different structures relatively easily. We again observe that previous methods will eventually diverge while training over 10,000 steps. In particular, RNNPROP achieves significant progress in the early stage of the optimization, but the earlier pre-processing techniques do not help the meta-optimizer stay close to

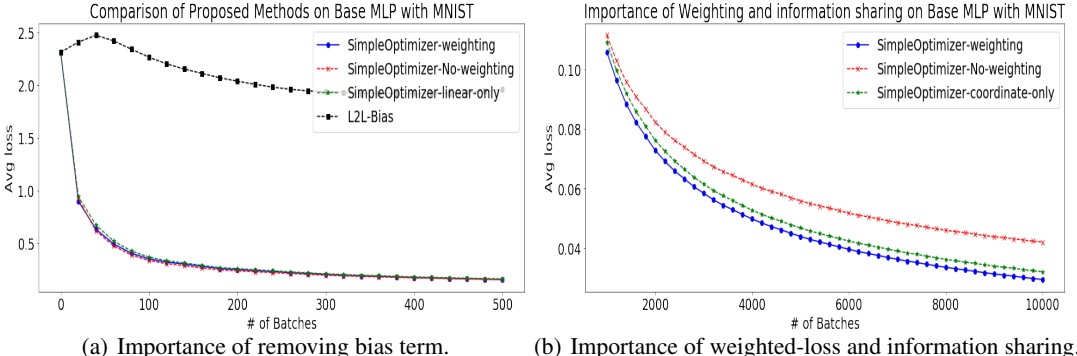

(a) Importance of removing bias term.     (b) Importance of weighted-loss and information sharing.

Figure 3: Ablation analysis of proposed SimpleOptimizer. (a) Adding back the bias term, we observe that meta-optimizer cannot converge. (b) With weighted difference of loss as the training objective, meta-optimizer further improves. Adding aggregate information sharing also helps in convergence.

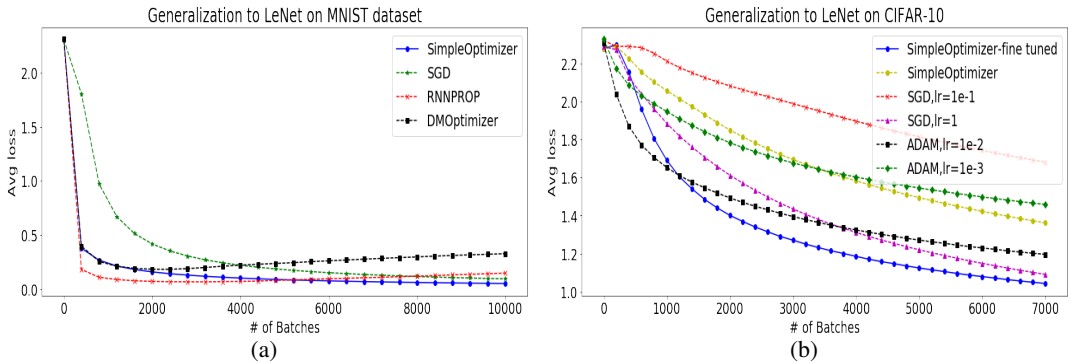

Figure 4: Generalization to different models and datasets. (a) Results of generalization to LeNet on MNIST dataset. (b) Results of generalization to LeNet on CIFAR-10 dataset.

the optimal solution. In the subsequent experiments, we focus on comparisons with standard gradient methods instead of meta-learning baselines.

**Generalization to CIFAR-10 dataset**    Now we generalize the base meta-optimizer to a new model structure and dataset at the same time. Results are shown in the Figure 4(b). Noticeably, SimpleOptimizer could achieve comparable results to not well-tuned traditional optimizer even without any tuning. This shows the good generalization capability of our method. When the fast fine-tuning procedure is applied, SimpleOptimizer converges faster than well-tuned SGD or ADAM.

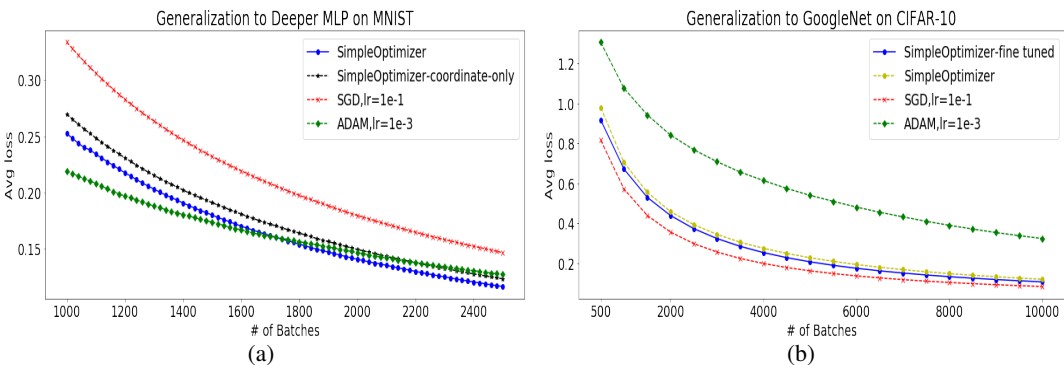

Figure 5: Generalization to deep models. (a) Experimental results of generalization to Deeper MLP on MNIST dataset. (b) Experimental results of generalization to GoogleNet V1 on CIFAR-10 dataset.

**Generalization to Deeper Models** In previous experiments, all models tested are relatively small. This is important to demonstrate the generalization capability of SimpleOptimizer on larger models. In this experiment, we aim to test the performance of SimpleOptimizer on larger models.

Firstly, we evaluate on 5 layers MLP with 16 times more parameters. Results are shown in the Figure 5(a) Surprisingly, SimpleOptimizer without any fine-tuning outperforms well-tuned traditional optimizers. This result implies that if the model structure is similar (in this case it is exactly the same except larger), learned optimizer can be directly applied without any fine-tuning.

Secondly, we apply the learned SimpleOptimizer to GoogleNet inception V1, which is 250 times larger than base MLP. Results are shown in Figure 5(b). It again shows that SimpleOptimizer is able to optimize the new model, and after the fast fine-tuning procedure, it can converge faster. However, SimpleOptimizer only outperforms ADAM but not well-tuned SGD.

These results basically validate the generalization capability of our proposed SimpleOptimizer. In particular, SimpleOpimizer does not diverge when the training steps is increased to 10k. In most cases we observe that SimpleOptimizer might not achieve the best performance in the early stage of the optimization, but eventually the loss drops faster than the traditional optimizers, demonstrating the non-myopic nature of our meta-optimizer. Also, fast fine-tuning works in all the cases to further improve the performance of the meta-optimizer. These experimental results validate the contributions of this work.

## 5 CONCLUSION

In this paper, we propose a simple yet effective meta-optimizer which can generalize to different structures and outperform well-tuned traditional optimizer. We analyze why the structure and loss used in previous meta-optimizers cannot lead to convergence in a long run, and propose simple fix for each of the components. Furthermore, we propose a novel information sharing scheme to further improve the convergence.

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
