# OpenReview forum: "LEARNING  TO LEARN  WITH  BETTER  CONVERGENCE"
_ICLR.cc/2020/Conference — Reject_

### Official Review · AnonReviewer1 · 2019-10-22
**Official Blind Review #1**

**Rating:** 3

**Review:**


This paper presents several improvements over the existing learning to learn models including Andrychowicz et al. (2016) and Lv et al. (2017). Specifically, this paper analyzes the issues in the original learning to learn paradigm (L2L), including instability during training and bias term issues in the RNN. It proposes a new loss based on weighted difference for improving the meta-optimizer in the later stage of optimization. It also proposes information sharing between RNNs for each coordinate. Finally it presents how to fine-tune the meta-optimizer on new task.


Pros:
Reasonable technical improvements to fix some issues of the learning to learn framework
(1) reduce the number of parameters in the meta-optimizer significantly
(2) improve the stability of the meta-optimizer.
(3) improve the generalization to new tasks and datasets.


Cons
1. The novelty is not good enough and the method does not seem to be solid enough. Except for the technique in the *structure of each recurrent unit" section, other techniques are tricks that are hard to tell why they could work. That said, I think the experiments should verify each of the proposed components, and see their roles in the proposed method.

Also, it is claimed in the Abstract and the paper that the proposed method *successfully converge to optimal solutions"... I think this claim is quite unprofessional. How do you know it converges the optima?


2. With respect to the experiments:

The experiments are only done on the MNIST and CIFAR10 datasets, which are small-scale datasets. Current meta learning model has achieved advancement on more challenging datasets, e.g., Mini-Imagenet and CIFAR1000.

For the experiment section of comparison to baseline methods. It would be fair to compare all the methods with removing bias or not removing bias setting. As the author mentions that "the maximal bias term of LSTM in Lv et al. (2017) is 0.41, and maximal bias term of LSTM in Andrychowicz et al. (2016) is 0.31; this, consequently, leads to bad performance". It would be interesting to know that if the bias term of the two baseline model are removed, how is the performance difference compared to the method proposed by the authors?

How does the number of parameters of the meta optimizer scales with the problem size? That is, how does the number of parameters in the meta-optimizer grow with increasing the number of image classes in the problem at hand, e.g., 20 classes, 50 classes, etc?

**Experience Assessment:**

I have read many papers in this area.

**Review Assessment: Checking Correctness Of Derivations And Theory:**

I assessed the sensibility of the derivations and theory.

**Review Assessment: Checking Correctness Of Experiments:**

I assessed the sensibility of the experiments.

**Review Assessment: Thoroughness In Paper Reading:**

I read the paper at least twice and used my best judgement in assessing the paper.

---

### Official Review · AnonReviewer2 · 2019-10-23
**Official Blind Review #2**

**Rating:** 3

**Review:**

In this paper, the authors build on the 'learning to learn' work, that aims to leverage deep learning models with optimization algorithms, most commonly with recurrent networks.  The goal is to utilize meta-learners that can adapt the optimization update strategy based on data/experience.  The authors aim to tackle problems that often arise in such meta-optimization schemes, such as covergence and generalization problems.  The paper is overall well-written, and several experiments are presented

Building on previous work, the authors propose some variations to the meta-learning schemes and architecture.  I comment on these below.

- The authors remove the bias term based on the intuition that for the meta-learners, removing the bias can lead to better convergence.  This is supported by a set of experiments in Fig 3a (but with only one learning rate?).  This experiment shows an extraordinary difference in accuracy between including and not including the RNN bias.  I think, because this difference is substantial (along with papers that do use the bias in meta-learners showing good results overall), more evidence/ablations are required.

Can we can be sure that if the bias is non-zero (as in other reported works), we should expect a worse performance?.  Clearly this change seems to speed up the process, and in the example explained in the paper it makes sense, but could there be examples where lack of bias might lead to worse results?

- The authors claim that one of the problems of learning-to-learn algorithms is the late-stage optimization strategy.  While previous works introduce a weighting for the meta-optimizer's loss function (usually binary), the authors extend this to continuous monotonically increasing functions in order to weight the late stage steps of the algorithm more.  The authors propose compensating small loss difference happening during late optimization with a larger weight.   This naturally brings the question of which function should be used, as showing that any function will do seems difficult in practice. Have the authors tried any functions that failed or led to overfitting in comparison to other works?  Could one argue that if the loss difference is not large then perhaps the network can be more prone to overfitting at least in some cases?

Finally, the authors introduce embedding sharing between coordinates (aggregating hidden states), and also propose fine-tuning (sampling 5% of training data 5 times for warm-up).  Sharing hidden state information is expected to be useful (and has been often employed in literature), similarly to fine-tuning.  Also, 5% of dataset and 5 times seems to be an arbitrary choice - probably more related to the experimental section rather than the method itself.

a question on fine-tuning:  if fine-tuning was used for the proposed method, have the compared methods also been pre-trained to provide fair comparisons?  This is particularly relevant to fig 4B where only the fine-tuned method slightly overperforms SGD.


**Experience Assessment:**

I have published one or two papers in this area.

**Review Assessment: Checking Correctness Of Derivations And Theory:**

I assessed the sensibility of the derivations and theory.

**Review Assessment: Checking Correctness Of Experiments:**

I assessed the sensibility of the experiments.

**Review Assessment: Thoroughness In Paper Reading:**

I read the paper thoroughly.

---

### Official Review · AnonReviewer3 · 2019-10-25
**Official Blind Review #3**

**Rating:** 1

**Review:**

This work suggests a host of  improvements and simplifications to the meta-learning approach of Andrychowicz et. al.  The authors have carefully analyzed weaknesses in previous work and I think their experiments do suggest that they have improved on them.  However, I would still recommend rejecting, as imo
1: the  absolute state of these approaches seems unpromising, and
2: the authors do not do a good job contextualizing how well the learned optimization performs compared to more standard methods.
 Each step used to train the meta-optimizer could be used instead to pick hyperparameters (choice of which "classical" optimizer, and its hyper-parameters, like lr etc.).  However, there is not really any discussion about the trade-offs between the time it takes to train the meta-optimizer and the results that would be obtained by hyper-parameter search.   Indeed, even with the tiny hyper-parameter search spaces the authors use, in most of their figures, one of ADAM or plain SGD does comparably or better than their method.

It is not clear if the figures represent train or test loss; the authors should report both.  Furthermore, imo, almost the figures are cut off too soon- the models all still seem to be learning.    What are the error rates at the cutoffs on test and train?

Finally, I wonder at the choice of test problems.  Why not pick something where one naturally will need to run an optimization many times (e.g. style transfer) rather than a toy problem like mnist or cifar?  Note that there are already other approaches (not learning based on learning gradient descent) that have been successful there.




**Experience Assessment:**

I have published one or two papers in this area.

**Review Assessment: Checking Correctness Of Derivations And Theory:**

I assessed the sensibility of the derivations and theory.

**Review Assessment: Checking Correctness Of Experiments:**

I assessed the sensibility of the experiments.

**Review Assessment: Thoroughness In Paper Reading:**

I read the paper at least twice and used my best judgement in assessing the paper.

---

### Decision · Program_Chairs · 2019-12-19

**Decision:**

Reject

**Comment:**

This paper proposes an improved (over Andrychowicz et al) meta-optimizer that tries to to learn better strategies for training deep machine learning models. The paper was reviewed by three experts, two of whom recommend Weak Reject and one who recommends Reject. The reviewers identify a number of significant concerns, including degree of novelty and contribution, connections to previous work, completeness of experiments, and comparisons to baselines. In light of these reviews and since the authors have unfortunately not provided a response to them, we cannot recommend accepting the paper.